# Two-Dimensional Discommensurations: An Extension to McMillan's Ginzburg–Landau Theory

**Lotte Mertens** [1,2] , **Jeroen van den Brink** [2,3] **and Jasper van Wezel** [1,*]

1   Institute for Theoretical Physics, University of Amsterdam, Science Park 904,
    1098 XH Amsterdam, The Netherlands; l.mertens@uva.nl
2   Institute for Theoretical Solid State Physics, IFW Dresden and Würzburg-Dresden Cluster of Excellence
    ct.qmat, Helmholtzstr. 20, 01069 Dresden, Germany
3   Institute for Theoretical Physics, Technische Universit Dresden, 01069 Dresden, Germany
*   Correspondence: vanwezel@uva.nl

**Abstract:** Charge density waves (CDWs) profoundly affect the electronic properties of materials and have an intricate interplay with other collective states, like superconductivity and magnetism. The well-known macroscopic Ginzburg–Landau theory stands out as a theoretical method for describing CDW phenomenology without requiring a microscopic description. In particular, it has been instrumental in understanding the emergence of domain structures in several CDW compounds, as well as the influence of critical fluctuations and the evolution towards or across lock-in transitions. In this context, McMillan's foundational work introduced discommensurations as the objects mediating the transition from commensurate to incommensurate CDWs, through an intermediate nearly commensurate phase characterised by an ordered array of phase slips. Here, we extended the simplified, effectively one-dimensional, setting of the original model to a fully two-dimensional analysis. We found exact and numerical solutions for several types of discommensuration patterns and provide a framework for consistently describing multi-component CDWs embedded in quasi-two-dimensional atomic lattices.

**Keywords:** charge density waves; Ginzburg-Landau theory; domain formation

## 1. Introduction

Various materials display phases with charge density waves: periodic modulation of electronics charge density among a crystalline atomic lattice in (static) wave-like patterns. The presence of CDWs impacts, among other things, the electronic and transport properties of materials. Furthermore, CDWs can influence the emergence of other collective states, such as superconductivity or magnetism [1–10]. Theoretical models capturing the qualitative physics of the CDW phase are well known both starting from a microscopic description, such as in the Peierls model [11,12], and in terms of macroscopic order parameter theories based on the Ginzburg–Landau paradigm [13–16]. Moreover, recent advances in experimental techniques and material synthesis have enabled the detailed exploration of CDWs in various material classes, leading to tunable properties and potential applications in areas like nanoscale electromechanics and energy storage [17–20].

Often, multiple charge ordered phases are present in the phase diagram of a single material. Generically, these go from the high-temperature metallic phase to an incommensurate CDW at lower temperature, which can turn into an ordered array of commensurate patches as it is cooled further, and finally, locks into the lattice to become a fully commensurate CDW at the lowest temperatures. Whether some or any of these states appear in the phase diagram of any particular materials depends on their detailed material properties.

The incommensurate charge density waves (IC-CDWs) exhibit a periodic charge density modulation that does not precisely match the underlying crystal lattice, in the sense that the wave vectors describing their atomic displacements are not linear combinations of

the pristine lattice vectors. The incommensurate wave vectors appearing in specific CDW materials typically arise from the interplay between nesting in the electronic band structure and the momentum dependence of electron–phonon coupling [21–28].

Upon cooling sufficiently, an IC-CDW may undergo a second transition into a commensurate charge density wave (C-CDW) phase. In this state, the CDW wave vector is a linear combination of lattice vectors. The "lock-in" of the CDW to the atomic lattice is favoured by the Coulomb interaction between the positively charged atomic cores and negatively charged electrons. Some materials also have a phase interpolating between the high-temperature IC-CDW and the low-temperature C-CDW, characterised by an ordered arrangement of commensurate patches separated by domain walls and topological defects. Between commensurate patches, either the phase or the amplitude of the CDW can vary, or both. The patches can have a variety of shapes and sizes, which generically depend on temperature and pressure.

The initial theoretical exploration of this intermediate phase was undertaken by McMillan in the context of $2H$-TaSe$_2$, laying the groundwork for understanding its phenomenology [14,29,30]. In this work, McMillan introduced a model for an effectively one-dimensional crystal structure with a single charge density wave. His investigation showed that, within a specific temperature range, the formation of domain walls between commensurate patches (discommensurations) becomes favourable as a result of the balance between the contributions of the atomic lock-in energy and electron–phonon coupling. The original paper is widely cited and has been extended and applied to several materials, including higher harmonics or a position-dependent amplitude to model 2D discommensuration patterns [31–33]. Among others, this has been used to show that the introduction of a triple charge density wave in two dimensions reduces the IC-C phase transition in $2H$-TaSe$_2$ from a second- to a first-order transition [32]. Complementary to the extension of the CDW Ansatz in the earlier works, we here focused on the effect of the curl term in the free energy, which we considered using an exact minimisation of the free energy. This adapts McMillan's original Ansatz to general materials and, in particular, allowed us to explore the theory in more-realistic, higher-dimensional settings.

The material investigated in the original study, $2H$-TaSe$_2$, can be argued to be approximated by a combination of quasi-one-dimensional CDWs, because all CDW components align with high-symmetry directions in the atomic lattice. In contrast, materials like $1T$-TaSe$_2$ or $1T$-TaS$_2$ exhibit a CDW wave vector that is rotated with respect to the atomic lattice [34,35], necessitating a broader two-dimensional framework. Here, we will give a detailed derivation of McMillan's original results within a consistently two-dimensional theory. We will show that this leads to novel predictions for the orientation of discommensuration lines in generic CDW materials that are not captured in the simplified one-dimensional analysis.

## 2. Results

*Single-Q Free Energy*

A Ginzburg–Landau theory for charge order can be formulated for charge density modulations $\alpha = \sum_i Re(\psi_i)$. The CDW is then described by a sum of wave-like components of the form $\psi_i = \psi_0 e^{i(\phi_i + \vec{q}_i \cdot \vec{r})}$, where the amplitude $\psi_0$ serves as the order parameter for the multi-component CDW, while $\vec{q}_i$ and $\phi_i$ describe the wave vector and phase of the $i$th component. Spatial variations of $\psi_0$ and $\phi_i$ may be used to describe the formation of various types of domain walls.

The total free energy for a single-component (single-$Q$) charge density wave in two dimensions can then be written as [29]

$$F = \int d^2r \left[ \bar{a}\alpha^2 - b\alpha^3 + \bar{c}\alpha^4 + e|\vec{Q} \cdot (\vec{\nabla} - i\vec{Q})\psi|^2 \right.$$
$$\left. + f|\vec{Q} \times \vec{\nabla}\psi|^2 \right].$$

where $\vec{Q}$ is the preferred wave vector for the IC-CDW phase, which is determined by the electronic nesting or, more generally, by the momentum at which the full electronic susceptibility has a maximum [22,26–28]. The coupling constants $\bar{a}, b, \bar{c}, e$, and $f$ can be (and typically are) dependent on parameters like temperature or pressure. The terms proportional to $e$ and $f$ measure the energy cost of changing the CDW wave vector $q$ away from $\vec{Q}$. Here, $F_e$ (the term proportional to $e$) encodes the cost in energy of altering the wavelength, while $F_f$ is affected by rotations of the CDW wave vector.

The coupling constants can have spatial dependence as well, as long as they respect the lattice symmetries. This can be ensured by expanding them in terms of reciprocal lattice vectors $\vec{K}_i$ as $\bar{a}(\vec{r}) = \bar{a}_0 + \bar{a}_1 \sum_i e^{i\vec{K}_i^{(1)} \cdot \vec{r}} + \dots$ [29], where $\vec{K}_i^{(1)}$ denote the shortest possible reciprocal lattice vectors, $\vec{K}_i^{(2)}$ the second shortest, and so on. A similar expansion can be made for all coupling constants.

To reproduce the results of McMillan in describing the single-$Q$ IC-CDW and C-CDW phases in the quasi-two-dimensional material $2H$-TaSe$_2$, it suffices to include only the constant part of $\bar{a}, \bar{c}, e, f$, the terms up to $b_1$ in the expansion of $b$, and a constant CDW phase $\phi(r) = \phi$. All other terms either lead to higher-order effects or drop out of the analysis when performing the integral in the free energy expression. Keeping only these contributions, the free energy becomes:

$$F = \int d^2 r \Big[ \bar{a}_0 \psi_0^2 \cos^2(\phi + \vec{q} \cdot \vec{r}) + \bar{c}_0 \psi_0^4 \cos^4(\phi + \vec{q} \cdot \vec{r})$$
$$+ \Big( b_0 + 2b_1 \cos\big(\vec{K}^{(1)} \cdot \vec{r}\big) \Big) \psi_0^3 \cos^3(\phi + \vec{q}\vec{r})$$
$$+ e_0 \psi_0^2 |\vec{Q} \cdot (\vec{q} - \vec{Q})|^2 + f_0 \psi_0^2 |\vec{Q} \times \vec{q}|^2 \Big].$$

The spatial integrals over odd powers of periodic functions vanish, because of their cancelling positive and negative contributions. This can be used to also evaluate the integrals over even powers of a periodic function by using trigonometric addition formulae. As an explicit example, consider the $F_a$ term with the wave vector for its periodic function written as $\vec{q} = \frac{2\pi}{\lambda}(c_x, c_y, 0)$. Here, we take $c_x^2 + c_y^2 = 1$ such that $\lambda = 2\pi/|q|$ is the CDW's wavelength. We can then define a periodically repeated unit cell for the function $\cos^2(\phi + \vec{q} \cdot \vec{r})$ with edge lengths in the $x$ and $y$ directions equal to $\lambda/c_x$ and $\lambda/c_y$. The free energy density $\mathcal{F}_a$ for this term then becomes:

$$\mathcal{F}_a = \frac{c_x c_y}{\lambda^2} \int_0^{\lambda/c_x} \int_0^{\lambda/c_y} \bar{a}_0 \psi_0^2 \cos^2(\phi + \vec{q} \cdot \vec{r})\, dx\, dy$$
$$= \frac{c_x c_y}{\lambda^2} \int_0^{\lambda/c_x} \int_0^{\lambda/c_y} \bar{a}_0 \psi_0^2 \cos^2(\vec{q} \cdot \vec{r})\, dx\, dy$$
$$= \frac{c_x c_y}{\lambda^2} \int_0^{\lambda/c_x} \int_0^{\lambda/c_y} \bar{a}_0 \frac{\psi_0^2}{2}(1 + \cos(2\vec{q} \cdot \vec{r}))\, dx\, dy$$
$$= \frac{\psi_0^2 \bar{a}_0}{2}.$$

The shift introduced in the periodic function in the second line is made possible by the fact that we integrated over an entire unit cell of the periodically repeating pattern. The cosine in the third term contributes zero when integrated over due to its periodicity, and only the constant term in the third line is left.

The analysis can be repeatedly used to evaluate any of the integrals appearing in the Ginzburg–Landau theory. For the term $\mathcal{F}_c$, we use $\cos^4(z) = 3/8 + 1/2 \cos(2z) + 1/8 \cos(4z)$, and the only term surviving the integral is $3/8$, yielding $\mathcal{F}_c = 3\bar{c}_0 \psi_0^4/8$. For the elastic energy $\mathcal{F}_e$, we have

$$\mathcal{F}_e = \frac{1}{A} \int e_0 \psi_0^2 |\vec{Q} \cdot (\vec{q} - \vec{Q})|^2\, d^2 r.$$

Since the integrand is constant, this simply yields $\mathcal{F}_e = e_0 \psi_0^2 [Q_x(q_x - Q_x) + Q_y(q_y - Q_y)]^2$. Similarly, the term proportional to $f$ becomes $\mathcal{F}_f = f_0 \psi_0^2 |\vec{Q} \times \vec{q}|^2$. As the $b_0$ term is odd, it vanishes. The $b_1$ term, however, can give a non-zero contribution due to the lattice vectors $K_i$:

$$\mathcal{F}_{b1} = -\int 2b_1 \psi_0^3 \cos\left(\vec{K}^{(1)} \cdot \vec{r}\right) \cos^3(\phi + \vec{q} \cdot \vec{r}) \, d^2r$$
$$= -\frac{b_1 \psi_0^3}{4} \left(3\cos(\phi) \, \delta_{\vec{K}^{(1)}, \pm \vec{q}} + \cos(3\phi) \, \delta_{\vec{K}^{(1)}, \pm 3\vec{q}}\right).$$

Again, the integrals over all odd powers of the cosine vanish, *except* when the argument itself is zero. This happens when either $\vec{K}_i = \pm \vec{q}_i$ or $\vec{K}_i = \pm 3\vec{q}_i$ as the cosine can be expanded using $\cos^3(z) = \frac{1}{4}(3\cos(z) + \cos(3z))$. This $b_1$ term represents the lock-in energy coming from the Coulomb interaction between the atomic lattice and the electrons in the CDW.

Combining all terms gives the free energy density:

$$\mathcal{F} = \frac{\bar{a}_0 \psi_0^2}{2} + \frac{3\bar{c}_0 \psi_0^4}{8}$$
$$- \frac{b_1 \psi_0^3}{4} \left(3\cos(\phi)\delta_{\vec{K}^{(1)}, \pm \vec{q}} + \cos(3\phi)\delta_{\vec{K}^{(1)}, \pm 3\vec{q}}\right)$$
$$+ e_0 \psi_0^2 |\vec{Q} \cdot (\vec{q} - \vec{Q})|^2 + f_0 \psi_0^2 |\vec{Q} \times \vec{q}|^2.$$

The equilibrium CDW configuration will minimise the free energy with respect to the parameters $\psi_0$, $\vec{q}$, and $\phi$. In the $F_e$ and $F_f$ terms, the energy is minimised when the CDW wave vector $\vec{q}$ equals the preferred IC-CDW ("nesting") vector $\vec{Q}$. The $F_b$ term, however, is minimised when the CDW is commensurate with the atomic lattice, so that either $\vec{q} = \pm \vec{K}^{(1)}$ or $\vec{q} = \pm \vec{K}^{(1)}/3$. There are, thus, two competing processes, the lock-in with the lattice coming from the $b_1$ term and the nesting preference coming from the $e_0$ and $f_0$ terms. The interplay between these effects at different temperatures will determine the CDW phase diagram.

The $b_1$ term also determines the CDW phase $\phi$, since its contribution to the energy is minimised for $\phi = 2\pi m$ when $\vec{q} = \vec{K}^{(1)}$ and for $\phi = 2\pi m/3$ when $\vec{q} = \vec{K}^{(1)}/3$. In both cases, the preferred values of the phase are such that the electronic charge maxima in the CDW coincide with a lattice position. For incommensurate values of $\vec{q}$, the CDW phase does not influence the energy at all, as any shift of the CDW pattern leaves the combined CDW–lattice configuration invariant up to a redefinition of the origin.

## 3. Incommensurate Charge Density Wave

For incommensurate charge order within a two-dimensional atomic lattice, the CDW wave vector $\vec{q}$ equals the preferred "nesting" vector $\vec{Q}$. All of the terms $F_e$, $F_b$, and $F_f$ then vanish, and the free energy density needs to be minimised only with respect to the order parameter amplitude:

$$\partial_{\psi_0} \mathcal{F} = \psi_0 \left(\bar{a}_0 + \frac{3\bar{c}_0}{2} \psi_0^2\right) = 0.$$

Assuming that, to the lowest order in $T - T_c$, all temperature dependence is contained in the quadratic term, we can write $\bar{a}_0 = \bar{a}'(T - T_c)$ [13]. This yields two regimes. The first is for $T > T_c$, where $\psi_0 = 0$ and there is no charge order (disordered, metallic phase). The second regime with $T < T_c$ has $\psi_0 = \sqrt{\frac{-2\bar{a}_0}{3\bar{c}_0}}$ and contains the incommensurate CDW $\psi = \psi_0 e^{i(\phi - \vec{Q} \cdot \vec{r})}$.

For the sake of concreteness, we will consider this IC-CDW phase within a two-dimensional implementation of the model for $2H$-TaSe$_2$ studied by McMillan [14]. We, thus, introduce a hexagonal two-dimensional atomic lattice described by the lattice vectors:

$$\vec{a}_1 = a_0(\frac{\sqrt{3}}{2}, \frac{1}{2}, 0), \qquad\qquad \vec{a}_2 = a_0(-\frac{\sqrt{3}}{2}, \frac{1}{2}, 0).$$

This gives the reciprocal lattice vectors:

$$K_1 = \frac{2\pi}{a_0}(1/\sqrt{3}, 1, 0), \qquad\qquad K_2 = \frac{2\pi}{a_0}(-1/\sqrt{3}, 1, 0).$$

The three-component IC-CDW in McMillan's model for this material is assumed to align with the three high-symmetry directions of the atomic lattice, but the length of its wave vectors, $|Q|$, is observed to be 2% shorter than $|K^{(1)}/3|$ [29]. The IC-CDW wave vectors then become:

$$\vec{Q}_1 = \frac{\pi}{2.55a_0}(1, \sqrt{3}, 0), \tag{1}$$

with $\vec{Q}_1$, $\vec{Q}_2$, and $\vec{Q}_3$ related by three-fold rotations.

The charge modulation for the single IC-CDW component with wave vector $\vec{Q}_1$ is shown in Figure 1. As the IC-CDW does not repeat over any integer number of lattice points, the peaks in electron density indicated by black diagonal lines do not coincide with the lattice points (black), except for a single line in the lower left corner.

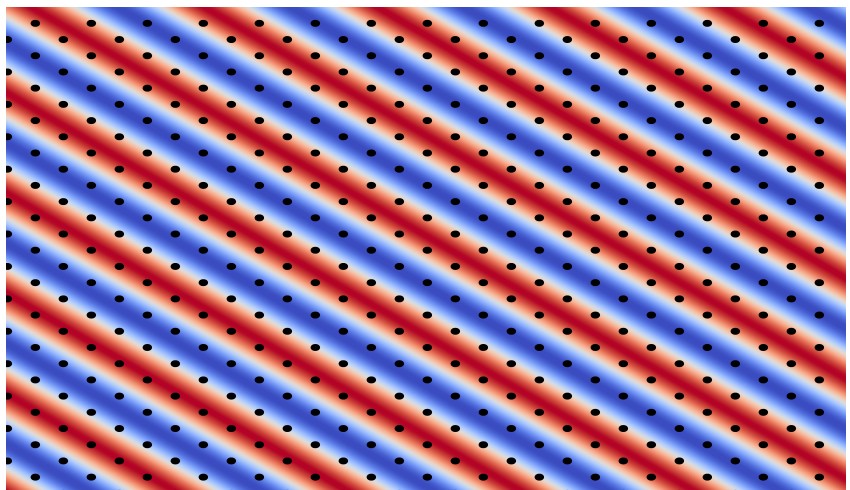

**Figure 1.** Incommensurate charge density wave with the wave vector of Equation (1). The colour scale indicates the electronic charge density modulation, ranging from $-\psi_0$ in blue to $+\psi_0$ in red. As the IC-CDW does not repeat over an integer number of lattice points, the peaks of the CDW do not coincide with the lattice points (black dots), except for a single line (lower left corner).

## 4. Commensurate Charge Density Wave

As temperature decreases, the amplitude of the order parameter $\psi_0$ increases, causing an increase in the contribution to $\mathcal{F}$ of the $b_1$ term relative to that of the $e_0$ term. Since the $b_1$ and $e_0$ terms favour different values of the wave vector $\vec{q}$, there may, thus, be a transition of the CDW wave vector away from the "nesting" vector $\vec{Q}$ at low temperatures. The energy cost due to the $e_0$ and $f_0$ terms encountered in a commensurate CDW is the lowest for the commensurate wave vector closest to $\vec{Q}$.

For $2H$-TaSe$_2$, the vector $\vec{C} = \vec{K}^{(1)}/3$ is the closest commensurate vector, the "nesting" vector $\vec{Q}$, with only a 2% difference in wavelength between the two. The charge density modulations for one of the components of this C-CDW is displayed in Figure 2 [29].

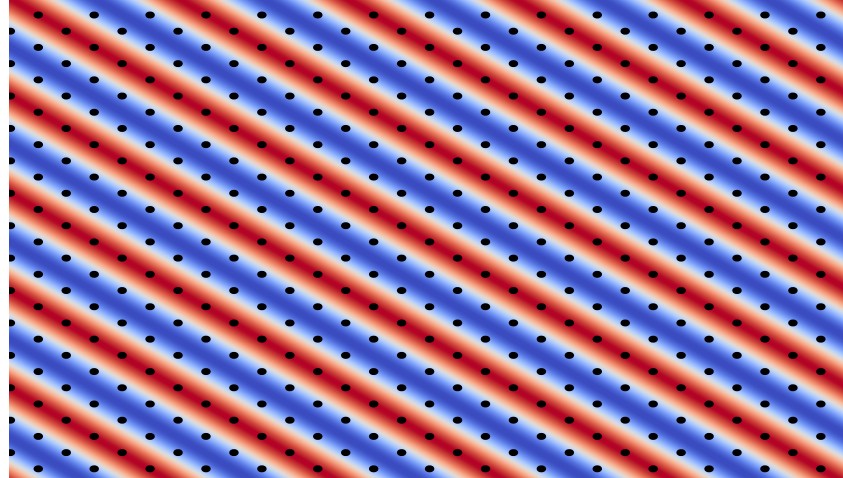

**Figure 2.** C-CDW with wave vector $\vec{C} = \vec{K}^{(1)}/3$. As the C-CDW repeats over a linear combination of lattice vectors, the ridges of CDW peak intensity $+\psi_0$ (red) always coincide with the rows of the lattice points (black dots).

Substituting the C-CDW Ansatz in the free energy, the equilibrium value of its amplitude $\psi_0$ and phase $\phi$ can again be determined by minimising the free energy. The minimisation with respect to the phase always yields the locked-in value $\phi = 0$. The amplitude, on the other hand, is temperature dependent, and found to be zero above the critical temperature $T_{c2} = T_c - \frac{2e_0}{\bar{a}'}(\vec{Q} \cdot (\vec{q} - \vec{Q}))^2 + \frac{b_1^2}{12c_0\bar{a}'}$. At the threshold, there is a first-order phase transition, and the amplitude jumps to:

$$\psi_0 = \frac{b_1}{4c_0} \tag{2}$$
$$+ \sqrt{\left(\frac{b_1}{4c_0}\right)^2 - \frac{2}{3c_0}\left(\bar{a}'(T - T_c) + 2e_0(\vec{Q} \cdot (\vec{q} - \vec{Q}))^2\right)}.$$

To determine whether the IC-CDW, C-CDW, or disordered phase will be energetically favourable at any given temperature, we can compare the free energy densities of their corresponding Ansatzs. In Figure 3, this is shown as a function of temperature for three different values of the parameter $b_1$ and (arbitrary) fixed values of the other parameters. At each temperature, the IC-CDW and C-CDW energies are shown for the value of $\psi_0$ that minimises the energy for the corresponding Ansatz. In the grey area at high temperature, it is not favourable for any CDW to form, and the phase is metallic ($T > T_c$). Going down in temperature, the second, blue area indicates the IC-CDW being the lowest-energy solution. Finally, the purple region shows the C-CDW with wave vector $\vec{K}^{(1)}/3$ being favoured. Depending on the value of $b_1 = 0.07$, the phase transitions separating these regions shift in temperature. Notice that, in this particular case, $\vec{K}^{(1)}/3$ and $\vec{Q}$ lie in the same direction such that $\mathcal{F}_f$ is zero regardless of the value of the $f_0$ parameter.

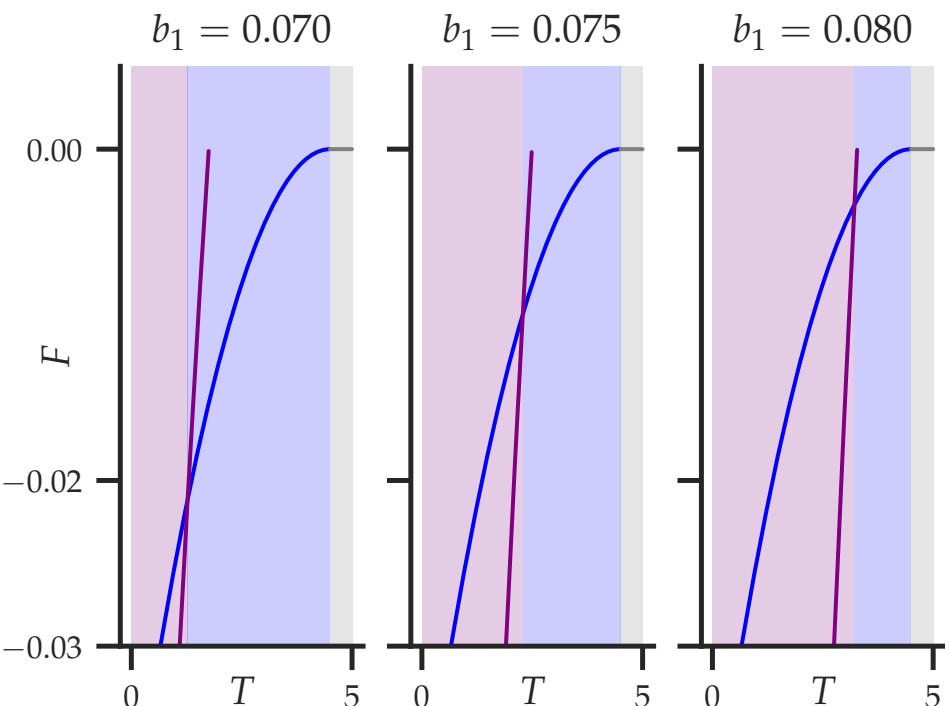

**Figure 3.** The energies of the IC-CDW, C-CDW, and disordered phases as a function of temperature. The blue line indicates the energy of the IC-CDW Ansatz with wave vector $\vec{Q}$. The purple line gives the energy of the C-CDW with wave vector $\vec{K}^{(1)}/3$. The grey area is the region where none of the CDW Ansatzs have an energy lower than zero, and the disordered, metallic phase is favoured. The blue area indicates the IC-CDW Ansatz having the lowest energy, and the purple area shows the C-CDW being favoured. Here, we used the parameter values $e_0 = 0.04$, $c_0 = 0.008$, $\bar{a}' = 0.01$, and $f_0 = 0$.

## 5. Discommensurations

So far, we have reproduced and given a pedagogical account of McMillan's description of the commensurate and incommensurate CDW phases in $2H$-TaSe$_2$ [29]. As shown by McMillan, however, there may also be an intervening phase between the IC-CDW and C-CDW phases, in which regions of commensurate CDW order are separated by lines of phase slips or discommensurations [29]. Within the Ginzburg–Landau theory, these defect lines can be included in the CDW order parameter $\psi$ by allowing the phase $\phi$ to be position-dependent. We, thus, consider the Ansatz:

$$\psi = \psi_0 e^{i(\phi(\vec{r}) - \vec{C} \cdot \vec{r})}, \tag{3}$$

such that, for $\phi$ being zero, the C-CDW with wave vector $\vec{C}$ is recovered, while for $\phi = (\vec{C} + \vec{Q}) \cdot \vec{r}$, the IC-CDW is recovered. Notice that, for the specific case of McMillan's model for $2H$-TaSe$_2$, the preferred commensurate wave vector is again given by $\vec{C} = \vec{K}^{(1)}/3$. Moreover, adding integer multiples of $2\pi/3$ to $\phi$ shifts the CDW pattern by an integer number of lattice distances, which does not influence the energy.

The free energy in the presence of a position-dependent phase can again be considered term by term. For general $\phi(\vec{r})$, the contribution proportional to $\bar{a}_0$ becomes:

$$F_a = \int \frac{\psi_0^2 \bar{a}_0}{4} \left(2 + \cos(2(\phi(\vec{r}) + \vec{q} \cdot \vec{r}))\right) d^2 r.$$

This integral cannot be evaluated exactly without specifying $\phi(\vec{r})$. For sufficiently smoothly varying functions $\phi(\vec{r})$, however, the second term in the integrand is approximately a plane wave everywhere. The integral over it, therefore, approximately vanishes,

and the contribution from the first, constant term dominates: $\mathcal{F}_a \approx \psi_0^2 a_0/2$. Similarly, we found for the quartic term that $\mathcal{F}_c \approx 3\psi_0^2 c_0/8$.

For the $b$ term, we have:

$$F_b = \frac{b_1\psi_0^3}{4}\int\left(\cos\left(3\phi(r) + 3\vec{q}\cdot\vec{r} + \vec{K}^{(1)}\cdot\vec{r}\right)\right. $$
$$\left. + 3\cos\left(\phi(r) + \vec{q}\cdot\vec{r} + \vec{K}^{(1)}\cdot\vec{r}\right)\right)d^2r.$$

The elastic energy term $F_e$ becomes:

$$F_e = \int e_0\psi_0^2\left(\vec{Q}\cdot(\vec{K}^{(1)}/3 - \vec{Q}) + \vec{Q}\cdot\vec{\nabla}\phi(r)\right)^2 d^2r.$$

Finally, the $F_f$ term can be written as:

$$F_f = \int f_0\psi_0^2\left(\vec{Q}\times\vec{K}^{(1)}/3 + \vec{Q}\times\vec{\nabla}\phi(r)\right)^2 d^2r.$$

To find the function $\phi(r)$ that minimises $F$, we need to take the two-dimensional functional derivative of the free energy and equate it to zero. This can be performed by first writing the free energy as:

$$F = \text{cst} + \int\left(-\frac{b_1\psi_0^3}{4}\cos(3\phi)\right.$$
$$\left. + e_0\psi_0^2(E_0 + \vec{Q}\cdot\vec{\nabla}\phi)^2 + f_0\psi_0^2(G_0 + \vec{Q}\times\vec{\nabla}\phi)^2\right)d^2r.$$

Here, "cst" is independent of $\phi$ and will, therefore, not contribute to the functional derivative. We also defined $E_0 = \vec{Q}\cdot(\vec{K}^{(1)}/3 - \vec{Q})$ and $G_0 = \vec{Q}\times\vec{K}^{(1)}/3$. Setting the functional derivative of $F$ with respect to $\phi$ equal to zero then yields:

$$\frac{3\psi_0 b_1}{4}\sin(3\phi)$$
$$= 2e_0(Q_x\partial_x + Q_y\partial_y)(E_0 + Q_x\partial_x\phi + Q_y\partial_y\phi)$$
$$+ 2f_0(-Q_y\partial_x + Q_x\partial_y)(G_0 + Q_x\partial_y\phi - Q_y\partial_x\phi)$$

Simplifying this expression yields the differential equation:

$$\frac{3\psi_0 b_1}{8}\sin(3\phi)$$
$$= e_0(Q_x\partial_x + Q_y\partial_y)^2\phi + f_0(Q_x\partial_y - Q_y\partial_x)^2\phi. \tag{4}$$

This expression can be recognised to be the differential equation describing the motion of a simple pendulum, which is solved by the Jacobi amplitude function:

$$\phi(x,y) = \frac{2}{3}\text{am}(c_1(x + Sy) + c_2, m) + \frac{\pi}{3},$$
$$\text{with } m = \frac{9\psi_0 b_1}{8c_1^2(e_0(Q_x + Q_y S)^2 + f_0(Q_x S - Q_y)^2}.$$

The full two-dimensional function is specified by the parameters $\psi_0$, $c_1$, $c_2$, and $S$. Among these, the integration constants $c_1$ and $c_2$ can be constrained by specifying boundary conditions on $\phi(x = 0, y = 0)$, as well as on $\partial_x\phi(x,y)|_{x=0,y=0}$ and $\partial_y\phi(y)|_{x=0,y=0}$. As a reminder, some of the properties and special values of the Jacobi amplitude function are:

$$\text{am}(x, 0) = x$$
$$\text{am}(x + c, 1) = \pi/2 \qquad\qquad \Longleftrightarrow \quad c \gg 1$$
$$\text{am}(x + 2K, m) = \text{am}(x, m) + \pi$$

Here, $K = \int_0^{\pi/2} d\theta / \sqrt{1 - m\sin^2(\theta)}$ is the quarter period.

For McMillan's model of $2H$-TaSe$_2$, the C-CDW phase is represented by $\phi = 2\pi n/3$ with $n \in \mathcal{Z}$. This solution can be written as a Jacobi amplitude function in terms of the limit:

$$c_2 \gg 1 \qquad\qquad c_1^2 = \frac{9\psi_0 b_1}{8(e_0 S_1^2 + f_0 S_2^2)}.$$

Here, $S_1 = Q^x + SQ^y$ and $S_2 = Q^x S - Q^y$, so that the function $\phi(x, y)$ does not depend on $S$ for the C-CDW. The IC-CDW phase can similarly be written as a Jacobi amplitude function by choosing:

$$S = \sqrt{3} \qquad\qquad \psi_0 b_1 = 0,$$
$$c_2 = -\frac{\pi}{2} \qquad\qquad c_1 = \frac{1}{2}(3Q_x - K_x^{(1)}).$$

The Jacobi amplitude function can also be used to interpolate between the IC-CDW and C-CDW Ansatzs. For general parameter values, it has approximately constant sections smoothly connected with steps of height $2\pi/3$ occurring every $2K$ (shown in Figure 4). This corresponds to a CDW Ansatz with commensurate patches separated by lines of phase shifts across which the CDW is moved by precisely one lattice distance in its propagation direction. These are the discommensurations that McMillan proposed for his model of $2H$-TaSe$_2$ [29]. The direction or slope of the discommensuration lines in the two-dimensional $x, y$ plane is determined by the value of the parameter $S$.

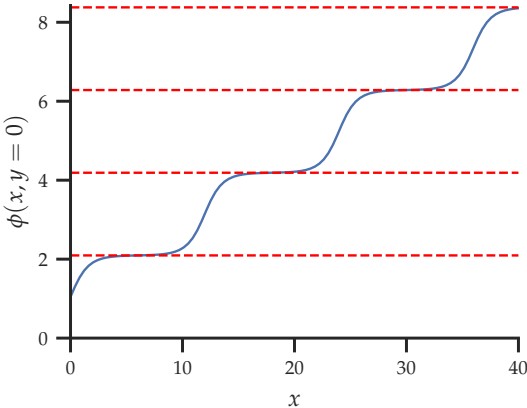

**Figure 4.** A slice of the Jacobi amplitude $\phi(x, y = 0) = 2/3\text{am}(ux, m) + \pi/2$ with $u = 1$ and $m = 0.9999$ as a function of position $x$. The function has steps whenever $x$ equals an integer multiple of $2K = 11.98$. The red dashed lines indicate integer multiples of $2\pi/3$ along the $y$-axis.

### 5.1. The Equilibrium Configuration

With any specific set of values for the coupling constants in the free energy, the values for $\psi_0$ and the parameters in the Jacobi amplitude function yielding the lowest possible free energy can be found using a numerical optimisation routine. The energy of the equilibrium configuration for $S = \sqrt{3}$, $\bar{a}' = 0.01$, $b_1 = 0.048$, $c_0 = 0.04$, $e_0 = 0.008$, and $T_c = 4.5$ is shown as a function of temperature in Figure 5 (red line). The value of $f_0$ is irrelevant as $S_2 = 0$ for $S = \sqrt{3}$. For each temperature, numerical optimisation using the Nelder–Mead algorithm is performed to find the parameter values that minimise the energy of the Ansatz based on the Jacobi amplitude function on a lattice of $300 \times 300$ sites. The energies of the IC-CDW (blue line) and C-CDW (purple line) Ansatzs are shown for comparison.

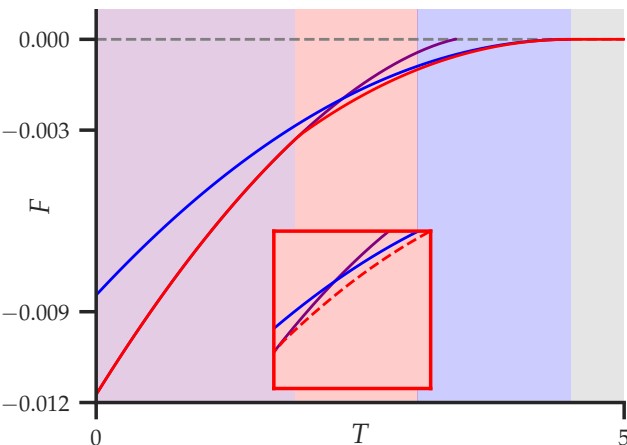

**Figure 5.** Free energies as a function of temperature for the IC-CDW Ansatz (blue line), the C-CDW Ansatz (purple line), and the discommensuration Ansatz based on the Jacobi amplitude function (red line). Here, we used $S = \sqrt{3}$, $\bar{a}' = 0.01$, $b_1 = 0.048$, $c_0 = 0.04$, $e_0 = 0.008$, and $T_c = 4.5$. For each temperature, numerical optimisation using the Nelder–Mead algorithm was performed to find the parameter values that minimise the energy of the Ansatz based on the Jacobi amplitude function on a lattice of $300 \times 300$ sites. The inset zooms in on the lines in the red area where the discommensuration Ansatz has significantly lower energy than the IC-CDW and C-CDW.

Between the phases where either the IC-CDW or the C-CDW has the lowest energy, we found a region where the discommensuration Ansatz using the Jacobi amplitude function with finite-sized domains has the overall lowest energy. The optimised functions of $\phi$ for different temperatures are displayed in Figure 6.

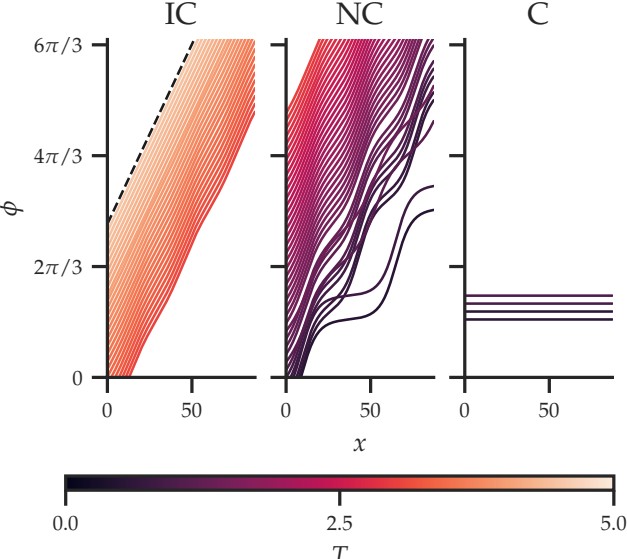

**Figure 6.** Slices of the phase $\phi(x, y = 0)$ as a function of position for different temperatures $T$, in three different CDW phases. All $\phi$ were vertically offset to separate the curves. The IC-CDW phase (left panel) has approximately no domain walls and becomes a straight line aligning with the exact IC-CDW Ansatz (dashed black line) at high temperatures. Lowering the temperature, the lowest-energy Ansatz crosses over into a regime with clear finite-sized discommensurations separating domains of finite width (middle panel). The phase slip across each of the discommensurations is $2\pi/3$. At even lower temperatures, the commensurate state obtains the lowest energy and $\phi$ becomes constant (right panel). The value of $\phi = 2\pi/3$ shown here is determined by the boundary conditions. The optimisation was performed on a lattice of $300 \times 300$ sites using $S = \sqrt{3}$, $\bar{a}' = 0.01$, $b_1 = 0.048$, $c_0 = 0.04$, $e_0 = 0.008$, and $T_c = 4.5$.

The Ansatz with the lowest energy is incommensurate for high temperatures, and $\phi$ is approximately a straight line, as is visible in the left of Figure 6. The dashed black line shows the exact IC-CDW Ansatz. The regime in the middle panel shows the discommensuration phase with a domain of around the same width as those found by McMillan [29]. In the right panel, the lowest-energy Ansatz is shown to be the C-CDW phase, in which $\phi$ is constant. Because the equilibrium configurations were determined using a numerical optimisation routine, the results can vary slightly depending on the initial conditions and the search algorithm used. The qualitative behaviour shown in Figure 6 was verified in multiple runs and with multiple choices for the initial conditions.

The parameter $c_1$ determines the width of the domain walls and domain interiors, while the constant $c_2$ only shifts the Jacobi amplitude function as a whole. In this Ansatz, the width of the domain wall and the domain's interior are thus co-dependent. The $c_1$ that minimises the free energy in the discommensuration phase is determined by the coupling constants coefficients $e_0$ and $b_1$, as well as $\psi_0$. The slices visible in Figure 6 are one-dimensional cuts through a two-dimensional structure. The fill two-dimensional pattern contain stripe-like domains, with the domain walls perpendicular to the CDW propagation vector due to the choice of $S = \sqrt{3}$, as shown for one particular choice of the parameters in Figure 7. Any parallel one-dimensional cuts taken through the two-dimensional (infinitely large) structure are equivalent, rendering the problem effectively one-dimensional.

### 5.2. Rotation in Two Dimensions

To observe the full effect of the CDW being embedded in two dimensions, we can release the constraint on $S$ and minimise the free energy for $S$, as well as the other parameters in the discommensuration Ansatz. This allows the orientation of the domain walls to vary away from being perpendicular to the CDW propagation vector. An example of the resulting discommensuration pattern is visualised in the bottom panel of Figure 7 for $S = \sqrt{(3)} - 1$. This construction allows for the generalisation of McMillan's Ansatz to truly two-dimensional discommensuration configurations.

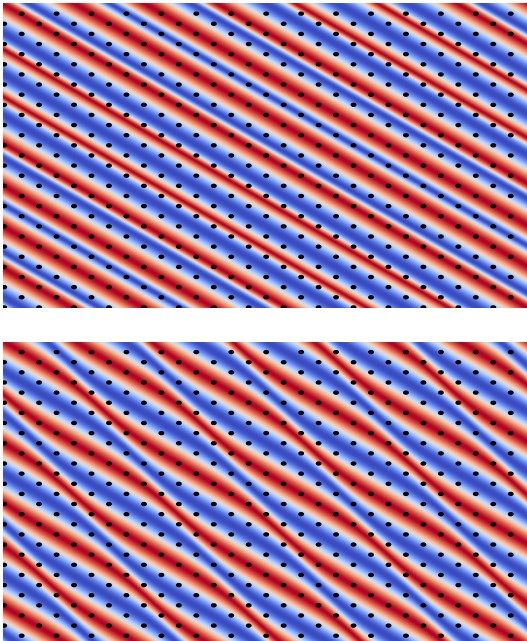

**Figure 7.** Electronic charge density modulations in the discommensuration phase. Here, we used $S = \sqrt{(3)}$ for the top panel and $S = \sqrt{(3)} - 1$ for the bottom. In both panels, we used $c_1 = 8$, $c_2 = 0$, and $m = 0.9999$. The colour denotes the amplitude of the charge modulations, ranging from $-\psi_0$ in blue to $\psi_0$ in red.

The energy of the two-dimensional discommensuration Ansatz can be minimised with respect to $S$, $c_1$, $c_2$, and $\psi_0$ on a lattice of $200 \times 200$ sites. This gives the patterns shown for different temperatures in Figure 8 as the equilibrium configurations. The left panel displays the one-dimensional slice $\phi(x, y = 0)$ for different temperatures. The right panel indicates the orientation $S$ of the domain walls as a function of temperature.

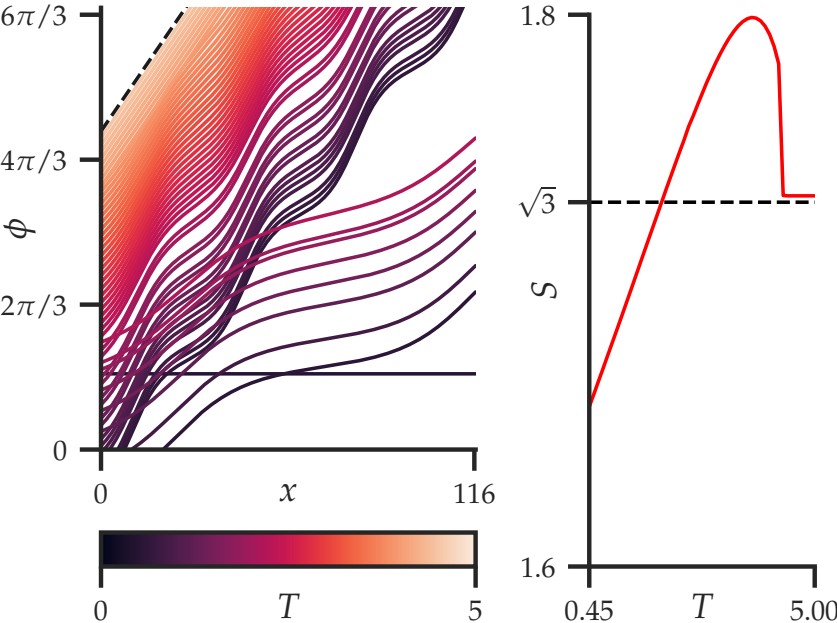

**Figure 8.** Left panel: Slices of the phase $\phi(x, y = 0)$ as a function of the position for different temperatures $T$, in three different CDW phases. All $\phi$ have been vertically offset to separate the curves. At low temperatures, the C-CDW with a constant phase is found to have the lowest energy. At high temperatures, the phase becomes a straight line matching the IC-CDW Ansatz shown as a dashed black line. At intermediate temperatures, an Ansatz with clear discommensurations is most favourable. Right panel: The orientation $S$ of the domain walls in the Ansatz with lowest energy, as a function of temperature. The horizontal dashed black line is the value $S = \sqrt{3}$ for which domain walls appear perpendicular to the CDW wave vector. Here, we used $\bar{a}' = 0.01$, $c_0 = 0.04$, $e_0 = 0.0008$, $f_0 = 0.002$, and $T_c = 4.5$.

At high temperatures, the lowest-energy Ansatz approaches the IC-CDW solution, and the slope of the domain walls is found to be $S = \sqrt{3}$, indicating the domain walls are perpendicular to the CDW wave vector. For the low-temperature C-CDW phase, in which $\phi$ is constant and there is only a single domain, $S$ loses meaning, and the temperatures in which the C-CDW Ansatz has the lowest energy are omitted from the right panel of Figure 8. In the discommensuration phase favoured at intermediate temperatures, the optimal value for the slope $S$ was found to vary between 1.65 and 1.8, surrounding the value $S = \sqrt{3}$. The variation of $S$ was confirmed not to originate in numerical artefacts by establishing its stability under changing initial conditions. The two-dimensional electronic density modulations for the configuration obtained when $S$ has its lowest equilibrium value of 1.65 is displayed in Figure 9. The small absolute value of the difference between 1.65 and $\sqrt{3} \approx 1.73$ implies that the difference between Figures 7 and 9 is hard to see with the naked eye within the limited field of view. Following the thinnest blue region in Figure 9 from the top to the bottom of the figure, however, a small oscillation around the lattice sites can be distinguished.

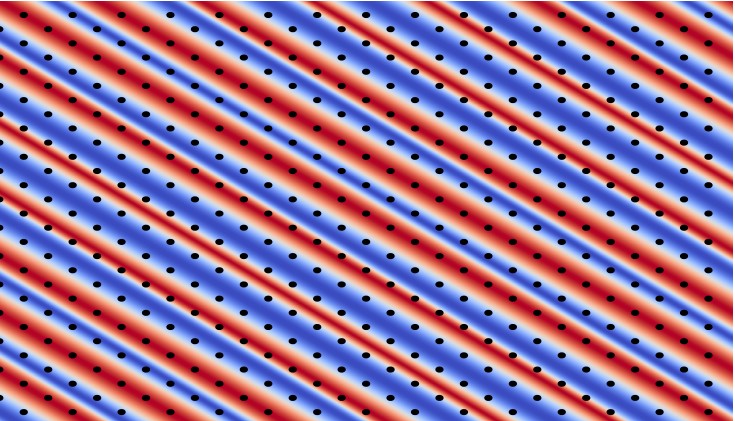

**Figure 9.** Electronic charge density modulations in the discommensuration phase. Here, we used the parameter values obtained from the energy minimisation at $T = 0.45$, which were found to be: $S = 1.65$, $c_1 = 8$, $c_2 = -\pi/2$, and $m = 0.9999$. The colour denotes the amplitude of the charge modulations, ranging from $-\psi_0$ in blue to $\psi_0$ in red.

The energy cost associated with the variation of the CDW phase across domain walls originates from the $F_e$ and $F_f$ terms in the free energy, while the energy gain of having the local C-CDW structure within the domains is provided by the $b_1$ term. Considering a regime in which the $b_1$ term is sufficiently dominant to favour the formation of discommensurations, a domain wall could reduce the cost of the $F_f$ term to zero by orienting itself perpendicular to the CDW wave vector. The $F_e$ term does cost energy in that case, because of the rapid variation of the phase across the domain wall, making the local wave length appear shorter than its preferred value. Starting from this situation, we can keep the width of the domain wall constant, but rotate it slightly so as not to be perpendicular to the CDW wave vector any longer. This will cost energy from the $F_f$ term, but since it stretches the effective local wave length observed within the domain wall, it reduces the $F_e$ cost. The reduction in the cost associated with $F_e$ generically scales linearly with the rotation angle, while the increase in $F_f$ will generically be quadratic (since it starts from an absolute minimum). We, thus, expect it to typically be favourable for $S$ to deviate slightly from the orientation perpendicular to the CDW wave vector, in agreement with the numerical results shown in Figure 8.

## 6. Conclusions

McMillan introduced discommensurations into the theory of charge density waves in his seminal work on the Ginzburg–Landau model for 2*H*-TaSe$_2$ [29]. This model showed that it can be favourable for a charge density wave to create commensurate domains separated by discommensurations rather than switching directly from a fully incommensurate to a fully commensurate phase. The original treatment was of an effectively one-dimensional model for a two-dimensional material. Here, we gave a detailed derivation of these original results in a consistently two-dimensional setting and went beyond them by also allowing for intrinsically two-dimensional discommensuration patterns and specifically the effect of the curl in the free energy. The orientation of domain walls in the two-dimensional configuration is governed by the competition between the lock-in effect, the preferred orientation of local charge density modulations, and their preferred local wave length. We showed that, as a result of this competition, discommensuration lines in two-dimensional CDW materials will rotate away from being perpendicular to the CDW wave vector. Even though the expected rotation angle will typically be small, the effect is predicted to occur generically. When the direction of the incommensurate wave vector diverts further from the commensurate one, such as occurs for example in 1*T*-TaSe$_2$ or 1*T*-TaS$_2$, the rotation angle of domain walls may be expected to increase accordingly. The current results, thus, lay a basis for the consistent modelling of discommensurations in quasi-two-dimensional materials in general, including in particular within the family of transition metal dichalcogenides.

**Author Contributions:** Conceptualization, L.M., J.v.d.B. and J.v.W.; formal analysis, L.M.; writing L.M., J.v.d.B. and J.v.W. All authors have read and agreed to the published version of the manuscript.

**Funding:** This research received no external funding.

**Data Availability Statement:** Data available from the authors upon reasonable request.

**Conflicts of Interest:** The authors declare no conflict of interest.

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
