# Peer review of "Two-Dimensional Discommensurations: An Extension to McMillan’s Ginzburg–Landau Theory"

_condensedmatter, doi:10.3390/condmat8040100_

Round 1
Reviewer 1 Report
Comments and Suggestions for Authors
The work expands on a well-known paper by McMillan from the 1970s in describing discommensurations in CDW systems.
The topic is important and very timely, and the results are potentially of interest to the wider community.
However, I have some reservations regarding the fact that the introduction discusses systems on triangular layered Van der Waals lattices, but later in the paper a tetragonal lattice is discussed, and only stripes are shown, with no other, more complex domain shapes.
The second reservation concerns the fact that since McMillan there are notable works by Nakanishi and Shiba which discuss discommensurations in the Landau formalism, including the materials that are mentioned in the introduction. There are also papers by Villain and Bak who deal with discommensurations in a generic way.
It would be important to discuss the present work in the context of the above-mentioned literature.
Author Response
Below, we reply to the points raised by the referee one by one, including their comments in italic.
———
RESPONSE TO THE FIRST REFEREE
The work expands on a well-known paper by McMillan from the 1970s in describing discommensurations in CDW systems.
The topic is important and very timely, and the results are potentially of interest to the wider community.
We are grateful to the referee for taking the time to carefully read our manuscript and acknowledging the relevance of the topic.
However, I have some reservations regarding the fact that the introduction discusses systems on triangular layered Van der Waals lattices, but later in the paper a tetragonal lattice is discussed, and only stripes are shown, with no other, more complex domain shapes.
We thank the referee for pointing out the lack of clarity about the lattice used in our work. In fact, we only consider a triangular lattice throughout the manuscript. The lattices in the figures of the original manuscript appeared tetragonal because of a plotting error, but the actual calculation was done on the correct triangular lattice. We correct the geometry of the figures in the revised manuscript, and thank the referee for making us aware of this issue.
We agree with the referee that including more complex domain shapes would be interesting. However, the point of the current conference proceedings is to introduce a (different) extension to McMillan’s work. The stripes and material considered are used to illustrate this method and therefore serve as pedagogical examples for these conference proceedings. We would like to note that we are currently working on more involved applications of the method introduced here. The results of those studies go far beyond the scope of the current work, however, and will be published separately.
The second reservation concerns the fact that since McMillan there are notable works by Nakanishi and Shiba which discuss discommensurations in the Landau formalism, including the materials that are mentioned in the introduction. There are also papers by Villain and Bak who deal with discommensurations in a generic way.
It would be important to discuss the present work in the context of the above-mentioned literature.
We thank the referee for naming these important works, and include a discussion of their relation to the current work in the introduction of the revised manuscript. The current results complement the systematic extensions of the CDW Ansatz by Nakanishi & Shiba and Villain & Bak, by focusing on the effect that the curl term in the free energy has on the original, simplified Ansatz of McMillan’s.
Reviewer 2 Report
Comments and Suggestions for Authors
The paper titled “Two-dimensional Discommensurations: an extension to McMillan's Ginzburg-Landau Theory” by Lotte Mertens, Jeroen van den Brink and Jasper van Wezel submitted to Condensed Matter. discusses the role of the Charge density waves (CDW) in superconducting materials which plays a key role in the electronic properties of complex materials. The authors have extended the original 1D model to fully 2D case, considering several types of discommensuration patterns. In this way they present a framework for consistently describing multi-component CDW embedded in quasi-two-dimensional atomic lattices.The paper is original, and it covers a hot topic therefore it is worth of publication in Condensed matter.
Anyway, the paper needs to be improved by referring to 1) classical important works on the commensurate phases, incommensurate phases [Bak, P. (1982). Commensurate phases, incommensurate phases and the devil's staircase. Reports on Progress in Physics, 45(6), 587] and 2) recent experimental results on incommensurate and quasi-commensurate phases in Nickelates obtained by nanoprobes using synchrotron radiation [Campi, G., Bianconi, A., & Ricci, A. (2021). Nanoscale phase separation of incommensurate and quasi-commensurate spin stripes in low temperature spin glass of La2− xSrxNiO4. Condensed Matter, 6(4), 45].
Comments on the Quality of English LanguageMinor editing of English language required
Author Response
Below, we reply to the points raised by the referee one by one, including their comments in italic.
———
RESPONSE TO THE SECOND REFEREE
The paper titled “Two-dimensional Discommensurations: an extension to McMillan's Ginzburg-Landau Theory” by Lotte Mertens, Jeroen van den Brink and Jasper van Wezel submitted to Condensed Matter. discusses the role of the Charge density waves (CDW) in superconducting materials which plays a key role in the electronic properties of complex materials. The authors have extended the original 1D model to fully 2D case, considering several types of discommensuration patterns. In this way they present a framework for consistently describing multi-component CDW embedded in quasi-two-dimensional atomic lattices.The paper is original, and it covers a hot topic therefore it is worth of publication in Condensed matter.
We are grateful to the referee for taking the time to carefully read our manuscript and for their outspoken support of its publication.
Anyway, the paper needs to be improved by referring to 1) classical important works on the commensurate phases, incommensurate phases [Bak, P. (1982). Commensurate phases, incommensurate phases and the devil's staircase. Reports on Progress in Physics, 45(6), 587]
and 2) recent experimental results on incommensurate and quasi-commensurate phases in Nickelates obtained by nanoprobes using synchrotron radiation [Campi, G., Bianconi, A., & Ricci, A. (2021). Nanoscale phase separation of incommensurate and quasi-commensurate spin stripes in low temperature spin glass of La2− xSrxNiO4. Condensed Matter, 6(4), 45].
We thank the referee for naming these important works, and include them in the revised manuscript.
Reviewer 3 Report
Comments and Suggestions for Authors
The submission extends the McMillan formulations in 2D to describe the formation of discommensurate phases in charge density wave materials beyond quasi-1D. The work is solid and original and will inspire further theoretical and experimental studies. We recommend the paper for publication, but after more discussions.
The major drawback of the paper is that while it clearly explained the original McMillan formulation and current developments as well as the numerical procedures, a deeper physical insight is lacking. Specifically, the paper can be improved e.g.,
- In discussing the key differences in the results from an 1D vs. 2D formulation. We understand that the 2D CDW are harder to nest and more difficult to stabilize without the participation of the lattice. It would be helpful if the authors can comment on whether / how 2D makes it easier or harder to form discommensurate CDW phases.
- There are other materials such as 1T-TaS2. How would the current formulations apply to those materials, especially when Q gets very far from K/3?
We also remind the authors of a relevant experimental study of the IC vs. C-CDW in 2H-TaSe2 by Ph. Leininger, et al. in Phys. Rev. B 83, 233101 (2011). It would be interesting to see how the experimental observations can add support to the theoretical descriptions.
We note there is a typo where Fig. 5 was cited as Fig. 6 the first time the Fig. 6 was mentioned.
Author Response
Below, we reply to the points raised by the referee one by one, including their comments in italic.
———
RESPONSE TO THE THIRD REFEREE
The submission extends the McMillan formulations in 2D to describe the formation of discommensurate phases in charge density wave materials beyond quasi-1D. The work is solid and original and will inspire further theoretical and experimental studies. We recommend the paper for publication, but after more discussions.
We are grateful to the referee for taking the time to carefully read our manuscript and for their recommendation for publication.
The major drawback of the paper is that while it clearly explained the original McMillan formulation and current developments as well as the numerical procedures, a deeper physical insight is lacking.
We agree with the referee that the current conference proceeding does not reveal any deeper physical understanding of recent experimental results. Instead, it is intended as the presentation of a general method, explained in the context of a well-known and already understood example. We are currently working on more involved applications of the method introduced here. The results of those studies go far beyond the scope of the current work, however, and will be published separately.
Specifically, the paper can be improved e.g.,
In discussing the key differences in the results from an 1D vs. 2D formulation. We understand that the 2D CDW are harder to nest and more difficult to stabilize without the participation of the lattice. It would be helpful if the authors can comment on whether / how 2D makes it easier or harder to form discommensurate CDW phases.
We thank the referee for the suggestion. We agree with the referee that 2D materials tend to avoid nesting and CDW formation as compared to 1D systems. In the present work, however, the presence of a diverging electronic susceptibility (whether caused by nesting or with the assistance of the lattice) is assumed from the outset. Indeed, assuming the existence of a phase transition is the usual starting point for any Ginzburg-Landau theory, and in the present case the inverse electronic susceptibility chi^-1(q) can be recognized as the Fourier transform of the coefficients appearing in front of the quadratic terms in the free energy. It crosses zero at the phase transition.
The present work thus cannot be used to say anything about whether discommensurate CDW are easier to form in 2D than in 1D. Instead, the main difference between 1D and 2D highlighted in the current work, is the effect of the curl term in the free energy. In 1D this term does not exist, and it is almost always neglected also in higher dimensions. Here, we show that this term does have a physical effect in 2D, and that it can be systematically taken into account using the methodology presented in our manuscript.
There are other materials such as 1T-TaS2. How would the current formulations apply to those materials, especially when Q gets very far from K/3?
We thank the referee for pointing out this important question. We tried to capture this answer in the last paragraph of the conclusion, without going into details that could easily take a follow-up paper to discuss. We predict, and are working on a paper showing it, that the rotation will increase when Q has a different direction than K. We added some sentences to the conclusion elaborating on this.
We also remind the authors of a relevant experimental study of the IC vs. C-CDW in 2H-TaSe2 by Ph. Leininger, et al. in Phys. Rev. B 83, 233101 (2011). It would be interesting to see how the experimental observations can add support to the theoretical descriptions.
We thank the referee for mentioning this work, which we cite in the revised manuscript. The application of our model to the detailed multi-domain, multi-Q setting of this particular study, however, goes beyond the scope of the current conference proceedings, and we reserve it for a future publication.
We note there is a typo where Fig. 5 was cited as Fig. 6 the first time the Fig. 6 was mentioned.
We thank the referee for alerting us to this typo and corrected it in the revised manuscript.
Round 2
Reviewer 3 Report
Comments and Suggestions for Authors
The authors have addressed my comments. I recommend the paper for publication.